# Probiotic-Based Intervention in the Treatment of Ulcerative Colitis: Conventional and New Approaches

**DOI:** 10.3390/biomedicines10092236

**Published:** 2022-09-09

**Authors:** Jana Štofilová, Monika Kvaková, Anna Kamlárová, Emília Hijová, Izabela Bertková, Zuzana Guľašová

**Affiliations:** Center of Clinical and Preclinical Research MEDIPARK, Faculty of Medicine, Pavol Jozef Safarik University in Kosice, Trieda SNP 1, 040 11 Kosice, Slovakia; monika.kvakova@upjs.sk (M.K.); anna.kamlarova@upjs.sk (A.K.); emilia.hijova@upjs.sk (E.H.); izabela.bertkova@upjs.sk (I.B.); zuzana.gulasova@upjs.sk (Z.G.)

**Keywords:** ulcerative colitis, gut microbiota, probiotics, next-generation probiotics, postbiotics, fecal microbiota transplantation

## Abstract

Although there are number of available therapies for ulcerative colitis (UC), many patients are unresponsive to these treatments or experience secondary failure during treatment. Thus, the development of new therapies or alternative strategies with minimal side effects is inevitable. Strategies targeting dysbiosis of gut microbiota have been tested in the management of UC due to the unquestionable role of gut microbiota in the etiology of UC. Advanced molecular analyses of gut microbiomes revealed evident dysbiosis in UC patients, characterized by a reduced biodiversity of commensal microbiota. Administration of conventional probiotic strains is a commonly applied approach in the management of the disease to modify the gut microbiome, improve intestinal barrier integrity and function, and maintain a balanced immune response. However, conventional probiotics do not always provide the expected health benefits to a patient. Their benefits vary significantly, depending on the type and stage of the disease and the strain and dose of the probiotics administered. Their mechanism of action is also strain-dependent. Recently, new candidates for potential next-generation probiotics have been discovered. This could bring to light new approaches in the restoration of microbiome homeostasis and in UC treatment in a targeted manner. The aim of this paper is to provide an updated review on the current options of probiotic-based therapies, highlight the effective conventional probiotic strains, and outline the future possibilities of next-generation probiotic and postbiotic supplementation and fecal microbiota transplantation in the management of UC.

## 1. Introduction

Inflammatory bowel disease (IBD) has become a global disease with accelerating incidence in newly industrialized countries whose societies have become more westernized, lately. The incidence and prevalence of IBD are increasing worldwide, and nowadays approximately 0.2% of the European population suffers from IBD [1,2]. Ulcerative colitis (UC) and Crohn disease (CD) represent two major variants of IBD. Histopathological signs of UC include continuous superficial inflammation of the mucous membrane, extending from the rectum to the more proximal part of the colon. Inflammatory lesions in UC are localized exclusively on the mucosa, and the damage does not penetrate the muscle layer [3]. Although the pathogenesis of UC remains unclear, environmental factors in association with the host genetics, dysregulated immune system, and microbial dysbiosis are important elements in UC development [4]. Genetic predisposition plays a crucial role in the etiopathology of IBD; furthermore, the inheritable component seems to be stronger in CD than in UC [5]. Based on the information above, it can be assumed that environmental factors are key in promoting intestinal inflammation, especially through their impact on the microbiota composition. The composition of gut microbial communities varies significantly between healthy individuals and UC patients, which supports the role of the microbiota in the etiology of UC [6]. Nevertheless, the question remains unanswered as to whether the gut dysbiosis associated with imbalanced immune responses is the cause or the consequence of UC [5,7].

Despite the number of commonly available therapies, many patients are unresponsive to these treatments or experience secondary failure during treatment [8]. Hence, the development of new therapies and the discovery of alternative strategies targeting microbial dysbiosis are needed in the management of UC. Probiotics have been applied in the management of the disease to modify the gut microbiota, improve intestinal barrier integrity and function, and maintain a balanced immune response [9]. Nevertheless, conventional probiotics do not always provide the expected health benefits to the patient, so next-generation probiotics (NGP), fecal microbiota transplantation (FMT), and postbiotics therapies targeting dysbiosis in UC have been developed recently. This article provides an updated review on the current possibilities of probiotic-based therapies, highlights effective conventional probiotic strains, and outlines the future possibilities of next-generation probiotic and postbiotic supplementation and fecal microbiota transplantation in the management of UC.

## 2. The Role of Gut Microbiota in UC Pathogenesis

The gut microbiota has an irreplaceable position in the host physiology, as it ensures several metabolic, trophic, and protective functions. Advances in next-generation sequencing technology have significantly contributed to the current understanding of the involvement of gut microbiota in intestinal inflammation. Moreover, the impact of the gut microbiota on host physiology extends beyond the gut and seems to exert profound effects on mood, motivation, and higher cognitive functions [10]. Multi-omic analysis performed in the frame of Human Microbiome Project revealed that patients with both UC and CD have compositionally and functionally disturbed microbial balance of the gut microbiome during the active state of the disease [11,12,13]. The role of the gut microbiota in the development of IBD was revealed by animal studies using germ-free animals, which did not develop intestinal inflammation in germ-free conditions, but inflammation occurred after colonization of their gastrointestinal tract with some commensal bacteria [14]. Colonization of the gut of the conventional animal models by microbiota derived from patients with UC can also exacerbate intestinal inflammation and UC symptoms [15]. Moreover, the data demonstrated a characteristic increase in facultative anaerobes at the expense of obligate anaerobes, as well as molecular disruptions in microbial transcription and metabolites due to mucosal inflammation. The proliferation of some species at the expense of others may lead to a change of metabolism function and a different profile of microbial metabolites, such as changes in short-chain fatty acids (SCFA) and tryptophan levels, as confirmed in both UC and CD patients [11,16,17]. The maintenance of the state of gut eubiosis, characterized by high diversity and richness of the microbial community and harmonic microbial metabolism, plays a crucial role in the pathogenesis of UC. As a result of the synergy of environmental stimuli with the host’s genetic predisposition, microbial homeostasis is disrupted, a pro-inflammatory cascade is induced, and dysbiosis occurs. The bidirectional interactions between the involvement of the gut microbiome in the pathophysiology and treatment of UC are schematically shown in Figure 1.

### 2.1. Gut Microbiota Alterations in UC

The microbial imbalance in development and course of UC is generally characterized by reduced biodiversity and richness of the commensal microorganisms and an increase in the number of certain pathobionts, as supported by several clinical examinations and observations [18,19,20,21]. Although it has not yet been confirmed that the development of UC is associated with a specific bacterium, the available evidence shows that a reduction in the diversity of fecal microbiota, especially in the Firmicutes and Bacteroidetes phyla, is the most consistent indicator of UC. Reduced diversity of the gut microbiota composition, as well as certain microbes, is also associated with the later clinical course of UC in terms of the relapse rate during the remission period and intractability during the active period [20]. Interestingly, *Faecalibacterium prausnitzii*, which belongs to Firmicutes, is significantly reduced, while Proteobacteria and Actinobacteria are usually elevated in active UC [22]. Patients in relapse are characterized by a lower proportion of Clostridiales and a higher proportion of Bacteroidetes [23]. The number of SCFA-producing bacteria such as *Clostridium butyricum* and *F. prausnitzii* was reported to be decreased in patients with UC, consequently affecting the differentiation and expansion of Treg cells as well as the growth of epithelial cells [24]. A significant inverse correlation between disease activity and the count of butyrate-producing bacteria such as *F. prausnitzii* and *Roseburia hominis* was confirmed. Although Varela et al. [25] reported that the recovery of the *F. prausnitzii* population after relapse is associated with the maintenance of clinical remission, UC patients with quiescent disease still have reduced overall counts of the mentioned species [26]. Several studies also showed a link between UC and reduced abundance of *Roseburia* species and *Akkermansia muciniphila* [26,27]. The changes in gut microbiota composition are also associated with altered microbial metabolism. Sun et al. [28] compared the microbial metabolisms of healthy volunteers to those of patients with active and inactive forms of UC. Analysis revealed several metabolites were affected in patients with UC, including trimethylamine N-oxide and sphingosine-1-phosphate, which were the most elevated metabolites that positively cross-correlated with the abundance of *Roseburia*, *Klebsiella*, *Escherichia*, and *Shigella.*

Although the presence and the role of the gut virome have been less studied, there is assumption that virome may play an important role in UC pathogenesis, as well. Substantial alterations of the mucosa virobiota with functional distortion were confirmed in UC. Typical features for UC mucosa are high abundance of DNA viruses (e.g., *Caudovirales* bacteriophages) and low *Caudovirales* diversity and richness. Bacteriophages’ functions associated with host bacteria fitness and pathogenicity are significantly enriched in UC mucosa [29]. Temperate phages infecting *Bacteroides uniformis* and *Bacteroides thetaiotaomicron* were over-represented in active UC patients in comparison with non-UC patients [30]. Similarly, as in the bacteriome, there are highly specific interindividual differences in phage communities. The limited understanding of phage biology and the microbiological aspect of disease means that additional in vivo and in vitro studies are required to elucidate the roles of gut phages in IBD development and treatment [31].

Apart from changes in bacterial and viral populations, reduced biodiversity in the fungal community was also observed in IBD patients. Specific differences in fungal composition exist between CD and UC; typically, an increased Basidiomycota/Ascomycota ratio, a decreased proportion of *Saccharomyces cerevisiae*, and an increased proportion of *Candida albicans* were associated with IBD [32].

### 2.2. Drug-Microbiota Interactions in UC Treatment

Medication administration is the first option in treatment of UC. The choice of treatment strategy depends on severity, localization, and the course of the disease. In addition to controlling and suppressing symptoms (inducing remission), medication can also be used to decrease the frequency of symptom flare-ups (maintaining remission) [33]. With proper treatment, periods of remission can be prolonged, and periods of symptom flare-ups can be reduced over time. Prescribed medications for UC mostly fall into the following categories: aminosalicylates (ASAs), corticosteroids, immunomodulators, antibiotics, biologic therapies, and small molecules. None of these drugs can cure the cause of UC, but basically, their role is to reduce inflammation and help patients to achieve and maintain remission. Initial treatment for people with mild to moderate UC begins with the use of ASAs (balsalazide, mesalazine, olsalazine, and sulfasalazine) due to their anti-inflammatory properties. Other medications such as corticosteroids, immunomodulators, and biologic agents are available for more severe cases of UC and non-responders to ASA treatment [34]. As with most medications, the benefits of therapy usually far outweigh the side effects. Nevertheless, the side effects of the treatment make the patient’s life uncomfortable, including headaches, abdominal pain, diarrhea, nausea, increased risk of infection, skin rash, and decreased kidney function.

Furthermore, accumulating evidence suggests an interaction between commonly used drugs and gut microbiota. The direct effect of antibiotics on gut microbiota composition is undisputed, but population-based studies have found a relationship between several groups of drugs, including drugs for UC, and changes in gut microbiota [35]. The gut microbiota also plays a role in drug metabolism and can affect drug availability, efficacy, and toxicity. Thus, the gut microbiota may influence the effectiveness of UC pharmacological therapy and predict whether individuals will respond to treatment or not [36,37]. The efficacy of sulfasalazine depends on its metabolism by the bacterial enzyme azoreductase, which reduces the prodrug to sulphapyridine and the active molecule 5-acetylsalicylic acid (5-ASA) [38]. The feces of germ-free animals treated with sulfasalazine contain an unmodified molecule of the drug, in contrast to conventional animals, suggesting a key role of the gut microbiota in ASA metabolism [39,40]. In addition, ASA treatment is more effective in patients with UC than in patients with CD because it is released exclusively in the colon, depending on microbial metabolism [41]. However, it has been shown that 5-ASA treatment affects the composition of the gut microbiota and is associated with elevated levels of some Firmicutes genera such as *Enterococcus*, *Lactobacillus*, and *Lactococcus* and decreased abundance of *F. prausnitzi*, *A. muciniphila*, *Bacteroides*, *Prevotella*, and some Proteobacteria such as *Escherichia* and *Shigella* in patients with UC [42]. Liu et al. [43] reported inhibitory effects of immunosuppressive drugs including azathioprine, mercaptopurine, and 5-ASA on the IBD-associated bacteria *Campylobacter concisus*, *Bacteroides fragilis*, and *Bacteroides vulgatus*. Furthermore, 5-ASA directly affected transcription of the virulence genes associated with the motility, adherence, and invasion of enteropathogenic *Escherichia coli* [44]. Like aminosalicylates, the active forms of corticosteroids are released from the prodrug in the intestine, and the process is strictly dependent on the activity of two bacterial enzymes: glycosidases and sulfatases [45]. The main targets of corticosteroids are to inhibit the inflammation and support epithelium healing [46]; however, some of them, such as dexamethasone and prednisolone, were reported to have modulatory effects on gut microbiota composition [47]. Conventional mice C57Bl/6 injected i.p. with dexamethasone showed a substantial shift in gut microbiota, with evident elevation of Actinobacteria, *Bifidobacterium*, and *Lactobacillus*, while the known colonic mucin degrader *Mucispirillum* was absent after 10 and 28 days of treatment. Atherly et al. [48] observed a different spatial distribution of mucosal bacteria in IBD dogs following prednisone therapy. Significantly higher numbers of *Bifidobacteria* and *Streptococci* were detected across all mucosal compartments, and elevated numbers of *Bifidobacterium* spp., *Faecalibacterium* spp., and *Streptococcus* spp. were present within the adherent mucus of IBD-diagnosed dogs after an 8-week prednisone treatment. The change in microbiota composition of IBD patients was documented also after biologic therapy, which can be associated with clinical response to the treatment. The main signs of gut microbiome changes in IBD patients receiving biologic treatment include an increment in SCFA-producing bacteria and a decrease in *Escherichia* and *Enteroccocus* [49]. The gut microbiota composition in patients treated with adalimumab (TNF-α inhibitor) shifted towards a microbiota typical for healthy individuals [50]. Moreover, differences in patient gut microbiota before and during biologic therapy may affect the period of remission. Sakurai et al. [51] have shown differences between the microbiota of UC patients who did and did not maintain remission after the discontinuation of adalimumab. Relatively higher abundance of the pro-inflammatory bacteria *Fusobacterium* sp. And *Veillonella dispar* at the beginning of treatment and relatively lower abundance of *Dorea* sp. and *Lachnospira* sp. After 24 weeks of treatment were associated with relapse, which occurred earlier than 72 weeks after the anti-TNF-α treatment’s initiation. Magnuson et al. [52] reported that responders and non-responders to TNF-α therapy displayed distinctly separate patterns of mucosal antimicrobial peptide (AMP) expression and gut microbiota before the beginning of the treatment. Characteristic nominators in responders were increased expression of defensin 5 and eosinophilic cationic protein, lower gut dysbiosis, and higher abundance of *F. prausnitzii*.

Based on the results above, it is evident that the gut microbiota has an ability to influence treatment outcomes in a drug–gut microbiota metabolism dependent manner. Therefore, more studies are required that focus on drug–gut microbiota interactions.

## 3. Conventional Probiotics and UC

Because the role of the gut microbiota has been confirmed in the etiology of UC, microbiota modulation is an attractive area for the application of approaches such as probiotic supplementation to alleviate inflammation and induce intestinal homeostasis. Twenty years have passed since the first definition of probiotics was established [53], and conventional probiotics have become a common part of prevention and therapy for a broad spectrum of gastrointestinal and non-gastrointestinal diseases. Probiotic therapy is commonly applied in the management of the diseases to modify the gut microbiota, improve intestinal barrier functions, and maintain a balanced immune response. The most common microorganisms used as conventional probiotics are lactobacilli, bifidobacteria, other bacteria such as *Streptococcus thermophilus* or *Escherichia coli* Nissle 1917, and certain yeasts (*Saccharomyces boulardii*). These conventional probiotics are very well characterized and marked as GRAS—generally recognized as safe [54]. Nevertheless, conventional probiotics do not always provide the expected health benefits to a patient. The biological effects of probiotics are strain- and dose-specific; therefore, the success or failure of one strain cannot be extrapolated to others. Furthermore, the probiotic benefit could vary depending on the type and stage of the disease as well as the host’s immune status.

### 3.1. Mechanisms of Action of Probiotics in UC

The precise mechanism of action of probiotics administered in clinical trials to patients with UC has rarely been investigated. Actual studies are focused more on monitoring the alleviating effect of probiotics throughout clinical manifestations of the disease. The exact mechanisms underlying the beneficial effects of individual probiotics are still under investigation and have been supported by preclinical studies employing various animal models of colitis [55,56,57,58,59]. Furthermore, it has been shown that the mechanism of action depends on the strain and dose used, as well as the severity of colitis [60,61]. The main target of probiotics is gut mucosa, where the probiotics adhere and interact with residing microbiota, epithelial cells, and gut immune system components. In UC, the gut mucosa overreacts to the presence of microorganisms and their antigens, followed by gut barrier disruption, increased permeability leading to excessive bacterial translocation and exposure of the host to luminal content, and overall exaggerated reaction of the mucosal immune system [62]. From this point of view, the mechanisms utilized by probiotic bacteria to mitigate UC progression can fall into two main categories: (1) those that affect the colonization and growth of pathobionts; and (2) effects associated with regulating over-activated immune responses and promoting barrier functions.

The beneficial changes in microbial composition observed after probiotic consumption in UC animal models are well documented [55,57,59,63]. In terms of microbial changes, probiotics use both the mechanism of competitive exclusion and inhibitory substance production [64]. Probiotics compete with pathogenic bacteria for adherence sites and nutrients; they also produce metabolites such as lactic acid and SCFA, resulting in reduced pH of the gastrointestinal tract, which makes it an unfriendly environment for pathogens. Furthermore, probiotics produce a broad spectrum of substances with direct anti-microbial effects, such as hydrogen peroxide, and species-specific bacteriocins [65]. Microbial changes are associated with altered gut metabolism [7], and administration of some probiotic strains has been documented to improve metabolic functions including amino acid, vitamin, and carbohydrate metabolism in mice with colitis [55,57]. Probiotic treatment may also beneficially affect the levels of anti-inflammatory metabolites (gamma-linolenic acid, carnosic acid) and antioxidants (ascorbic acid, 25,26-dihydroxyvitamin D) at the systemic level in UC [57]. Various strains of lactobacilli and bifidobacteria have been shown to significantly reduce the adhesion and invasion of adherent-invasive *E. coli* LF82 in HT29 intestinal epithelial cells. These probiotic strains also reduced the adhesion index of pathogenic bacteria to the abiotic surface in biofilm experiments, suggesting that they affect the expression of *E. coli* LF82 adhesion determinants rather than using a competitive mechanism for host cell receptor sites on epithelial cells [66].

The immunomodulatory activity of probiotics is exhibited through their interactions with epithelial and immune cells residing in the gut mucosa. The presence of probiotics is recorded via the innate immune pattern-recognition receptors, such as Toll-like receptors (TLR), expressed on both epithelial and antigen-presenting cells [67]. TLR signaling influences innate and adaptive immune components. The activation or inhibition of transcriptional factors involved in inflammatory processes occurs depending on the recognized probiotic strain [68]. Probiotics are able to attenuate inflammatory processes in UC by inhibiting the expression and production of pro-inflammatory transcription factors and/or molecules, and/or by inducing immunoregulatory mechanisms such as the differentiation of Th regulatory lymphocytes in the gut mucosa [69]. Several strains of *Limosilactobacillus* (formerly *Lactobacillus*) *fermentum* have been shown to mitigate inflammatory processes by inhibiting the expression of pro-inflammatory transcription factors (nuclear factor-κB p65, mitogen-activated protein kinases p38 and JNK1/2) and downregulating the level of pro-inflammatory molecules (inducible nitric oxide synthase, cyclooxygenase 2) and cytokines (IL-6, TNF-α, IL-12) [69,70,71]. Other probiotic strains, such as *Lactiplantibacillus* (formerly *Lactobacillus*) *plantarum* and *Bifidobacterium longum*, have been documented to alleviate chemically induced colitis by restoring the Th17/Treg balance in the lamina propria of the colon [72]. Probiotic bacteria might attenuate inflammation also by regulating the maturation of dendritic cells (DC) and producing tolerogenic DCs [73,74,75]. The immunomodulatory effect of probiotics is often coupled with their ability to improve the integrity and permeability of the disrupted intestinal barrier [76]. The protective effects of probiotics on the inflamed gut mucosa are manifested mainly through the upregulation of tight junction proteins (zonulins, claudins, and E-cadherins), the increase of mucin production, and the stimulation of regulatory cytokine IL-10 and reactive oxygen species-scavenging enzymes, such as superoxide dismutase, catalase, and glutathione peroxidase 2 [59,77,78,79]. In addition to enhancing tight junction strength, the strain *Lacticaseibacillus* (formerly *Lactobacillus*) *rhamnossus* GG was also reported to confer protection against oxidative stress-mediated apoptosis of epithelial cells [79].

### 3.2. Effectiveness of Conventional Probiotics in Clinical Trials

All studies cited in this paper assessed a probiotic formulation previously known as VSL#3. The formulation of the probiotic product VSL#3, which is currently marketed under this brand name, is not identical to the original product, which since 2016 has been known under the generic name ‘De Simone Formulation’ and is available on the market under the brand names Visbiome (USA) and Vivomixx (Europe).

Conventional probiotics have been tested in the treatment of human UC for a decade [80,81]. Several meta-analyses and reviews summarize the prophylactic and therapeutic potential of probiotic preparations in clinical trials [82,83,84,85,86]. Based on the clinical trial evidence available to date, it seems that patients with active UC profit mainly from treatment with *E. coli* Nissle 1917 (Mutaflor) and the probiotic mixture VSL#3 [87,88,89,90,91]. VSL#3 is a probiotic cocktail of eight live freeze-dried bacterial species comprising *Lacticaseibacillus* (formerly *Lactobacillus*) *casei*, *L. plantarum*, *Lactobacillus acidophilus*, *Lactobacillus delbrueckii* subsp. *bulgaricus*, *B. longum*, *Bifidobacterium breve*, *Bifidobacterium infantis*, and *Streptococcus salivarius* subsp. *thermophilus*. Administration of VSL#3 resulted in an induction of clinical remission and symptom mitigation in active UC, and there was no increased risk of adverse effects. In addition, more than half of patients with mild to moderate active UC who did not respond to conventional treatment recovered after 6 weeks of VSL#3 treatment [88]. Similarly, the non-pathogenic strain *E. coli* Nissle 1917 is generally accepted as being as effective and safe as mesalazine, the gold standard in maintaining remission in patients with UC [91]. In the guidelines of the European Crohn’s and Colitis Organization (ECCO), *E. coli* Nissle 1917 was acknowledged as an evidence-based medicinal substance belonging to the group of probiotics for maintaining the remission of UC in both adults and children [92]. Species of the *Lactobacillus* and *Bifidobacterium* genera are the ones most commonly used in UC therapy, and some studies even indicate that it is possible to replace medical treatment with probiotic supplementation. *L. rhamnossus* GG was more effective at prolonging relapse-free time than mesalazine [93]. The probiotic strain *B. longum* BB536 showed its beneficial effects after 24 weeks of supplementation by reducing the clinical activity index, inducing remission by improving the colonic mucosal condition and modulating the secretion of inflammatory cytokines in patients with mild to moderate UC that was refractory to conventional therapy [94]. Most of the studies suggest that probiotics alone are effective in the treatment of UC, especially when several strains are concomitantly administered [87,95,96,97,98,99,100,101]. Nevertheless, the clinical efficacy of conventional probiotics for inducing and maintaining remission of UC is limited, and there are studies in which some probiotic strains failed to induce remission or alleviate UC symptoms [100,101]. Interestingly, in both such studies, the probiotic strain *L. acidophilus* was applied to treat UC. Despite its failure, the strain is effectively used to treat acute, chronic, and antibiotic-associated diarrhea [102], confirming the importance of selecting the appropriate strain for a particular disease.

Because patients with active UC suffer from severe gastrointestinal problems, immediate medication intervention is unavoidable. Therefore, most clinical trials have examined the efficacy of probiotics as an adjuvant form of UC therapy in patients who were already being treated with mesalazine. VSL#3 supplementation after 8 weeks was able to reduce disease activity scores in patients affected by relapsing mild to moderate UC who were concomitantly under treatment with 5-ASA and/or immunosuppressants [87]. Groege et al. [103] reported a positive effect 6 weeks of *B. infantis* intake, observing reduced systemic pro-inflammatory biomarkers, CRP, and IL-6 in UC patients receiving mesalazine. In the study of Palumbo et al. [96], all UC patients treated with combination therapy (mesalazine + *Ligilactobacillus* (formerly *Lactobacillus*) *salivarius*, *L. acidophilus*, and *Bifidobacterium bifidus* BGN4) showed improvement in Mayo Disease Activity Index and reduced stool frequency, and endoscopic pictures showed an improvement in gut mucosa signs compared to controls, who received only mesalazine. However, the adjuvant effect of probiotics was more evident after two years of the treatment. The effectiveness of a probiotic strain depends on the disease to be treated and may even vary among diseases with a similar etiology, such as UC and CD. This was confirmed by Bjarnason et al. [98], who applied *L. rhamnossus* NCIMB30174, *L. plantarum* NCIMB 30173, *L. acidophillus* NCIMB 30175, and *Enterobacterium faecium* NCIMB30176 for 4 weeks in patients with asymptomatic IBD. Probiotics were able to reduce intestinal inflammation in patients with UC but not in those with CD. This may be due to different immune system responses among diseases, as Th1 skewing is observed in CD, Th2 skewing is characteristic of UC, and each probiotic strain is characterized by either pro- or anti-inflammatory potential. Interestingly, the method of probiotic administration also affects its efficacy. D’Inca et al. [104] reported that rectally administered *L. casei* DG modified colonic microbiota by increasing *Lactobacillus* spp., reducing *Enterobacteriaceae*, inhibiting TLR-4 and IL-1β gene expression, and significantly increasing the level of mucosal IL-10 in UC patients. However, these effects were not confirmed after oral administration of this strain at the same dosage.

In addition, probiotic therapy can be potentially improved by combining it with a prebiotic substance. Dietary prebiotics are defined as “a selectively fermented ingredients that results in specific changes in the composition and/or activity of the gastrointestinal microbiota, thus conferring benefit (s) upon host health” [105]. Prebiotics are usually non-digestible oligosaccharides that are not absorbed in the upper gut and selectively support the growth of indigenous beneficial bacteria in the colon. Moreover, their standalone administration can lead to the suppression of inflammation as well as changes in microbial composition and metabolism associated with increased SCFA production [106]. Some clinical trials have shown that the intake of synbiotics (probiotics together with prebiotics) seems to also be an effective method of UC management [97,99,107,108]. Reduced sigmoidoscopy scores and inflammation, the regeneration of epithelial tissue, and reduced expression of β-defensins and the pro-inflammatory cytokines TNF-α and IL-1α were documented after 6 weeks of synbiotic supplementation composed of *B. longum* and a fructooligosacharide/inulin mix in patients with active UC [107]. Altun et al. [97] reported improvements in clinical values and reduced CRP and sedimentation values after 8 weeks of supplementation with a probiotic mix (*Enterobacterium faecium*, *L. plantarum*, *S. thermophilus*, *B. lactis*, *L. acidophilus*, *B. longum*) and fructooligosaccharides in patients with mild to moderate UC. Similarly, Amiriani et al. [99] observed an improvement in clinical symptoms in patients with active UC after an 8-week intake of synbiotic Lactocare^®^. Patients with longer durations of active disease responded significantly better to the intervention (with Lactocare^®^) compared to those with shorter ones (less than 5 years). Furthermore, synbiotics with conventional medication as a combinational therapy appeared to be more effective in improving subjective symptoms such as quality of life, abdominal pain, and stool consistency [108]. A selection of recent clinical trials in which probiotics alone, in combination with prebiotics, or as an adjuvant therapy to 5-ASA have shown promising outcomes in UC, both active and in remission, are listed in Table 1.

## 4. Next Generation Probiotics and UC

Research in the field of probiotics has expanded to include microbes that do not fall under the umbrella of conventional probiotic species, which are referred to as “next-generation probiotics” (NGP). NGP are considered to be novel functional microbes with beneficial properties, and the term corresponds to newly isolated bacteria, mainly anaerobic ones [110]. In most cases, NGP are members of commensal microbiota belonging to diverse genera, apart from *Lactobacillus* spp. and *Bifidobacterium* spp., and they have been identified from the comparison of results of healthy and sick animals/humans [111]. NGP derived from the gut microbiota represent new preventive and therapeutic tools in the management of various diseases. There are many NGP candidates, such as *A. muciniphila*, *Christensenella minuta*, *F. prausnitzii*, *Clostridium butyricum*, non-toxigenic strains of *Bacteroides fragilis*, and *Anaerobutyricum soehngenii*, whose abundance has been affected in certain pathological conditions, including UC [112]. These microorganisms have all been studied in vitro and in preclinical conditions with an interest in identifying their mechanisms of action. In the next section, some of their beneficial properties targeting the pathology of UC are highlighted.

It has been confirmed that *A. muciniphila* is associated with healthy mucosa. However, its precise role in colitis is currently unknown. Bian et al. [113] confirmed that *A. muciniphila* treatment beneficially improved clinical signs of colitis (weight loss, colon length, histopathology score) in the DSS mouse model of colitis. Furthermore, *A. muciniphila* significantly inhibited serum and tissue levels of inflammatory cytokines and chemokines (TNF-α, IL1α, IL-6, IL-12a, macrophage inflammatory protein α (MIP-1α), eotaxin, granulocyte colony-stimulating factor (G-CSF), and keranocyte-derived chemokine) and induced a significant shift in gut microbiota composition associated with increased amounts of Verrucomicrobia, *Akkermansia*, *Ruminococcaceae*, and *Rikenellaceae*. *A. muciniphila* seems to be an irreplaceable member of healthy and balanced gut microbiota; furthermore, it alleviates mucosal inflammation either via microbe–host interactions associated with improved gut barrier function and reduction of inflammatory cytokines, or by improving the microbial community in UC. It is noteworthy that the beneficial effect of *A. muciniphila* was confirmed for both live and pasteurized forms used in different diseases, including liver injury [114], obesity, and diabetes 2 [115,116,117]. Ottman et al. [118] identified a highly abundant outer membrane pili-like protein, Amuc-100, of *A. muciniphila* MucT, which through the activation of TLR-2 and TLR-4 induces the production of the regulatory cytokine IL-10 and could be directly responsible for the anti-inflammatory properties of the bacterium. Similar to conventional probiotics, the latest results indicate that strain specificity matters also in the case of NGP application. Liu et al. [119] compared four strains of *A. muciniphila*, and only one of them was able to alleviate symptoms of UC in mice. The positive effect of *A. muciniphila* FSDLZ36M5 was associated with specific functional genes that are involved in immune defense mechanisms and protein synthesis. By contrast, it was shown that abundance of *A. muciniphila* is elevated in colorectal carcinoma (CRC), and studies in this field indicate that the bacterium could be involved in the pathogenesis of the disease and its progression [120,121]. This may be due to insufficient nutrition in patients with CRC combined with the natural ability of *A. muciniphila* to degrade mucin. Although *A. muciniphila* does not exhibit entero-invasive properties, excessive mucin degradation and thickening of the protective layer leave room for potential pathogens to overstimulate inflammatory mechanisms and trigger the progression of the disease [122]. Open clinical trials of *A. muciniphila* in humans have not yet been published, and therefore the safety of *A. muciniphila* in humans is still questionable.

Commensal bacteria with the ability to produce butyrate are also important members of the gut microbial community. Butyrate, the main energy fuel of colonocytes, ensures and maintains the gut barrier’s optimal, healthy condition via its trophic and anti-inflammatory activity. In addition, the D- and L-forms of lactate are end products of fermentation by primary degraders such as *Bifidobacterium* spp. and lactic acid bacteria, but they have been found to interfere with or promote the inflammatory response and/or adversely affect mucosal barrier function [123,124]. Nevertheless, lactate as well as acetate are important metabolites, which in the cross-feeding process are utilized by butyrate-producing bacteria [125]. *F. prausnitzii* is one of the most abundant bacterial butyrate producers found in the gut [126]. It was reported that its defective colonization and abundance are highly present in IBD patients, with more obvious depletion in CD than UC [127]. The recovery of the *F. prausnitzii* population after relapse is associated with the maintenance of clinical remission in UC [25]. The therapeutic potential of *F. prausnitzii* in UC has been confirmed by several animal experiments [128,129,130,131]. Intragastrically administered *F. prausnitzii* at a dose of 1 × 10^9^ CFU for 7 or 10 days alleviated the course and severity of colitis and supported recovery from dinitrobenzene sulfonic acid (DNBS)-induced colitis in mice. The lower severity of colitis was associated with the downregulation of myeloperoxidase, pro-inflammatory cytokines, and T-cell levels [128]. Zhou et al. [131] confirmed protective effect of *F. prausnitzii* and its metabolites against TNBS-induced colitis in mice. *F. prausnitzii* beneficially affected gut dysbiosis, which resulted in an increase in bacterial diversity and the abundance of SCFA-producing bacteria, a decreased level of serum TNF-α, and the abundance of Proteobacteria, Acidobacteria, and Bacteroidetes. Furthermore, *F. prausnitzii* improved the functioning of the intestinal epithelium in rats with DSS-induced colitis [129].

*Anaerobutyricum soehngenii*, an anaerobic bacterium belonging to the phylum Firmicutes, is capable of converting sugars as well as lactate and acetate into butyrate [132]. Cuffaro et al. [133] used an in vitro screening approach to highlight the beneficial properties of several commensal bacteria strains on human health, which are promising NGP candidates for the management of IBD and obesity. Among all the studied NGP candidates, *A. soehngenii* AS170 produced the most significant amount of butyrate, and it was also able to strengthen the epithelial barrier; however, it did not display immunomodulatory activity. Nevertheless, the authors found two specific strains, *Parabacteroides distasonis* AS93 and *Roseburia intestinalis* AS6, that were able to concomitantly induce IL-10 secretion, improve disturbed barrier function, and secrete GLP-1. Other appealing NGP candidates targeting IBD are *Bacteroides coprocola* AS101, *Bacteroides uniformis* PF-BaE8, and *Bacteroides uniformis* PF-BaE13, which combine an anti-inflammatory profile with the ability to improve the epithelial barrier.

Certain strains of *Clostridium butyricum* have been used for decades as probiotics, and their positive effect across several murine models of colitis has been confirmed [134,135,136,137]. Treatment with live *C. butyricum* CGMCC313.1 had a similar or better effect than mesalazine on levels of the inflammatory cytokines IL-23 and TNF-α, as well as on restoring the balance of the intestinal microbiota in rat model of colitis [134]. Xie et al. [138] confirmed a dose-dependent protective effect of *C. butyricum* on acute intestinal inflammation induced by DSS in mice via TLR2 signaling pathway inhibition, downregulation of IL-23 and RORγt expression, and inhibition of IL-17 secretion. Supplementation with *C. butyricum* CBM588 resulted in a protective effect in an acute DSS-induced colitis model, accompanied by an increase in IL-10 production in lamina propria mononuclear cells from the inflamed intestine [139].

Another potential NGP, *Christensenella minuta* DSM 22607, was demonstrated to possess strong anti-inflammatory activity, resulting in a decreased level of the pro-inflammatory cytokine IL-8 via NF-kB signaling pathway inhibition. Moreover, its anti-inflammatory activities were associated with improved intestinal epithelial functions and integrity in vitro. The same strain evidently prevented intestinal damage, reduced colonic inflammation, and promoted mucosal restoration in both TNBS- and DNBS-induced mouse models of colitis [140]. Furthermore, the non-pathogenic commensal strain *Bacteroides fragilis* has been shown to be a potentially effective NGP. *B. fragilis* mitigated mucosal inflammation via promoting the lineage differentiation of Foxp3+ Treg cells, mediated tolerance at mucosal surfaces through IL-10 production, and prevented intestinal inflammation [141].

The decreased abundance of commensal Gram-negative bacteria belonging to the *Bacteroides* species was negatively associated with UC [142]. Among them, *Bacteroides thetaiotaomicron* was identified as a potential NGP due to its important functions in nutrient absorption, anti-inflammatory effects, and promotion of barrier function. Furthermore *B. thetaiotaomicron* supplementation ameliorated DSS-induced colitis in rodents [143,144]. Germ-free mice colonized with the *B. thetaiotaomicron* presented enhanced expression of genes with intestinal barrier functions and did not display an increment in the expression of pro-inflammatory genes [144]. However, Durant et al. [145] have shown that outer membrane vesicles (OMVs) of *B. thetaiotaomicron* promote regulatory dendritic cell responses in healthy patients but not in UC patients in remission. OMVs were unable to elicit IL-10 expression by colonic DC or increase the levels of CD103+ DC in the colons of UC patients.

The growing body of knowledge about the composition and metabolic profile of gut microbiota in UC provides further opportunities to identify and select other potential NGP candidates. The findings on the beneficial administration of selected NGP in animal models of UC are the cornerstone for further clinical studies regarding their effective and safe application in patients with UC. Furthermore, future research focused on microbiome modulation in UC could also aim at the application of a combination of several NGP strains with previously confirmed efficacy in preclinical studies. These results suggest that NGP exhibit great potential to be novel agents in UC therapy.

## 5. Fecal Microbiota Transplantation and UC

Fecal microbiota transplantation (FMT) has been established as a life-saving approach in the treatment of recurrent *Clostridioides* (formerly *Clostridium) difficile* infection [146]. The confirmed involvement of the gut microbiota in the etiopathology of IBD and apparent dysbiosis have encouraged follow-up clinical research to focus on the use of FMT for both UC and CD treatment. The first attempts at FMT application in UC were aimed at restoring balance by correcting microbial dysbiosis while administering the microbiota from a healthy donor to a patient and inducing intestinal homeostasis associated with inflammatory process elimination to achieve remission. In previous studies which employed animal models of colitis, promising effectiveness of FMT was demonstrated for both microbiome restoration and inflammation suppression associated with improved clinical appearance [147,148,149]. The latest clinical trials focused on FMT have suggested a potential role of FMT in the treatment of mild to moderate UC [150]. Tian et al. [151] reported that FMT has potential therapeutic value for the treatment of UC as it beneficially affected the abundance of bacterial microbiota and improved scores for diarrhea, abdominal pain, and mucous membrane lesions in UC patients. In a case series published by Dang et al. [152], favorable clinical outcomes were achieved in 91.7% of patients with UC, and in two-thirds of such patients, clinical remission persisted after 52 weeks. Despite these positive results, the effectiveness of FMT treatment after relapse was reduced but sufficient to alleviate the severity of the disease in comparison with the initial state lacking FMT treatment. Crothers et al. [153] published a pilot study testing the enhanced effect of orally administered FMT capsules for 12 weeks in UC patients with previously delivered FMT via colonoscopic infusion. The authors observed that daily intake of encapsulated FMT may extend the durability of FMT-induced changes in gut bacterial community structure and induce remission. Even though the study partially confirmed the effectiveness of orally applied FMT, only 12 participants were involved, so the conclusions are questionable. Imdad and co-workers [154] also reviewed the use of FMT in UC patients and concluded that FMT helps to induce remission. Although FMT application initially appeared to be appropriate only in the quiescent phase of the disease due to sepsis elimination in patients with impaired intestinal barriers, several studies have confirmed that FMT is safe and effective also for active UC in patients who do not respond to mesalazine or prednisone [155]. FMT appears to have bright prospects in the long-term management of UC due to its multi-pronged attack on the pathophysiological mechanisms involved in the pathogenesis of UC [156]. The advantage of FMT is that the patient receives a complex and balanced microbiota from a healthy donor, which contains a spectrum of commensal bacteria, including the full range of potential NGP that are depleted or missing in the UC microbiome. Furthermore, some NGP such as *F. prausnitzii* may serve as diagnostic and therapeutic biomarkers for the use of FMT in UC [157]. Although FMT could bring about prolonged and faster clinical effects in comparison with probiotic administration, there are still contradictory results from different trials, and several issues need to be solved regarding donor selection, route of administration, dosage, therapy duration, standardization of treatment protocol, acceptability, and safety [158]. Nevertheless, confirmation of the same efficacy of FMT oral capsules as a fresh transplant would facilitate clinical application and increase the availability of the therapy for patients with UC. Even though several clinical trials have shown many positive aspects of FMT for patients with UC, further research is needed to establish that FMT is a proper treatment option in UC therapy [21,156,159,160,161].

## 6. Postbiotics and UC

There is growing evidence that the viability of microorganisms is not required in order to achieve the desired physiological effect on the host [162,163]. The health-promoting effects of non-viable probiotics and their various cellular parts or produced metabolites, known as postbiotics, are manifested due to their interactions with the host’s sensing cells exhibiting antimicrobial, antioxidant, anti-inflammatory, anti-proliferative, and immunomodulatory activities. A postbiotic is defined as a “preparation of inanimate microorganisms and/or their components that confers a health benefit on the host” [164,165]. Thus, postbiotics encompass diverse components of bacterial cells and their metabolites such as cell surface and other types of proteins, peptides, endo- and exo-polysaccharides, extracellular vesicles, SCFAs, bacteriocins, enzymes, teichoic acids, peptidoglycan-derived muropeptides, vitamins, plasmalogens, and organic acids or their mixtures known as cell-free supernatant (CFS) [166,167]. Because postbiotics are devoid of any viable microorganisms, they do not pose a risk of undesirable infection and therefore represent a safer therapeutic strategy, especially in immunocompromised patients. The advantage of postbiotics over viable probiotics is that they are more resistant to gastric stress and can be applied in a precise dose, so they are currently being developed as pharmacobiotics [168]. Although their mechanism of action has not yet been fully elucidated, they are thought to regulate host–microbiota crosstalk and affect cell pathways such as proliferation, differentiation, migration, and cell death. It was reported that the postbiotics improve mucosal maturation and function and, furthermore, trigger the host’s immune system and anti-inflammatory responses [169,170]. Postbiotics are important for maintaining intestinal homeostasis and epithelial barrier function, as well as for establishing stable communities within gut microbiota; thus, the use of these bioactive compounds in IBD therapy is recommended [171].

The most-studied postbiotics in IBD therapy are SCFAs, namely acetate, propionate, and butyrate, which are the main products of the microbial fermentation of dietary fibers in the colon. Lower levels of intestinal SCFAs were associated with dysbiosis, IBD, and furthermore, the occurrence of colorectal carcinoma [172,173]. SCFAs, mainly butyrate, a major energy source for colonocytes, are also responsible for the growth, proliferation, and protection from apoptosis of colonocytes, increasing mucin production, maintaining gut barrier integrity and function, and inhibiting inflammation and oxidative stress [174]. In contrast to normal cells, butyrate in cancer cells plays an opposite role, known as the “butyrate paradox”. Butyrate suppresses proliferation, supports differentiation, and evokes oxidative stress, leading to cell cycle arrest and apoptosis [175,176,177]. The concentration of butyrate is significantly lowered in IBD patients in comparison with healthy individuals, reflecting metabolic alterations likely caused by changes in the gut microbiota composition [178,179,180]. Typical butyrate-producing bacteria are *F. prausnitzii*, *Eubacterium rectale*, *C. butyricum*, *Butyricicoccus pullicaecorum*, *Bifidobacterium* spp., *Anaerostipes* spp., *Lachnospira* spp., and *Roseburia* spp. [181]. It has been shown that the cell-free supernatant (CFS) of *F. prausnitzii* exerts in vivo and in vitro anti-inflammatory activity. Indeed, CFS of *F. prausnitzii* downregulates the production of pro-inflammatory mediators such as TNF-α and IL-8 by blocking NF-κB activation and the IL-6/STAT3/IL-17 downstream pathway [182]. Furthermore, CFS has been shown to induce production of anti-inflammatory IL-10 by upregulating regulatory T-cell differentiation and inhibiting Th17 differentiation and IL-17A secretion in both DSS- and TNBS-induced colorectal colitis in rodents, which contributes to maintaining the balance of Th17/Treg cells [183,184]. CFS of *F. prausnitzii* has been reported to reduce the severity of acute, chronic, and low-grade chemical-induced inflammation in rodent models, and furthermore, it has been confirmed that the anti-inflammatory effects of *F. prausnitzii* in experimental UC are attributed mainly to butyrate, which also enhances intestinal barrier function and affects paracellular permeability [128,130,183]. Vernia et al. [185] showed that oral administration of butyrate enhanced the efficacy of oral mesalazine treatment in active UC. Sitkin et al. in a recent study showed that supplementation of mesalazine with butyrate in patients with active UC increased the butyryl-CoA:acetate-CoA transferase (BCoAT) gene content in fecal microbiota, reduced the elevated baseline *B. fragilis*/*F. prausnitzii* ratio, and improved disease symptoms [186]. Similarly, oral supplementation of microencapsulated sodium butyrate appears to be a valid adjuvant therapy for maintaining remission in UC patients [187]. In comparison with butyrate, Tedelind et al. studied the anti-inflammatory properties of other SCFAs, namely acetate and propionate. It was demonstrated that SCFAs inhibit lipopolysaccharide-stimulated TNF-α release, but not IL-8 secretion, from human blood-derived neutrophils. Furthermore, propionate and acetate, comparable to butyrate, dose-dependently inhibited TNFα-mediated activation of the NF-κB pathway in a human colon adenocarcinoma cell line (butyrate > propionate > acetate), and anti-inflammatory activity was demonstrated for acetate and propionate in an in vitro model of murine experimental colitis [188]. A growing body of evidence suggests that the SCFA supplementation is one of the key routes to improve the course of UC, and thus the introduction of probiotics based on SCFA-producing bacteria into clinical practice and SCFA supplementation may help to increase the effectiveness of conventional therapy in patients with IBD [131,189,190].

Heat-killed probiotic bacteria, CFS, and extracellular proteins were also identified as having the ability to reduce the pro-inflammatory response and inhibit cytokine-induced apoptosis in intestinal epithelial cells. Quevrain et al. [191] identified a 15 kDa protein produced by *F. prausnitzii* which was able to inhibit the NF-kB signaling pathway in intestinal epithelial cells and to prevent colitis in an animal model in a dose-dependent manner. Li et al. [192] demonstrated that the *L. rhamnosus* GG effector protein HM0539 had an inhibitory effect on the inflammatory response through the TLR4/MyD88/NF-кB axis signaling pathway. HM0539 induced a decrease in TLR4 expression and decreased MyD88 levels, leading to the inhibition of distal NF-κB activation and pro-inflammatory mediators, thereby attenuating LPS-induced inflammatory responses. Imaoka et al. [193] observed anti-inflammatory activity by two heat-killed probiotic bacterial strains, *B. breve* and *B. bifidum*, in fermented milk, which induced the secretion IL-10 production in peripheral blood mononuclear cells from UC patients and the inhibition of IL-8 secretion in HT-29 cells. Zagato et al. [194] showed in vitro and also in vivo that *L. paracasei* fermented products (both metabolic products in culture medium and fermented milk) could protect against colitis and against an enteric pathogen infection via the inhibition of inflammation. Research in this area opens new perspectives and suggests that postbiotics can provide health benefits without carrying live bacteria that may be potentially dangerous to IBD patients.

## 7. Conclusions

It is still unclear whether gut dysbiosis is a cause or a consequence of the chronic inflammation involved in IBD, including UC. Nevertheless, conventional probiotics appear to be a promising tool for the prevention and treatment of UC that targets both the deregulated immune response as well as gut dysbiosis and which is supported by a number of clinical trials. Probiotics help to maintain remission for longer periods and improve the quality of life of UC patients. However, it is important to carefully choose the right strain and time of application depending on the course of the disease. Probiotics such as *E. coli Nissle* 1917 and VSL#3 have been shown to be as effective in inducing remission in UC patients as the standard therapeutic drug—mesalazine. On the other hand, NGP, which represent depleted commensal bacteria in UC, have shown promising results on the outcomes of the disease in preclinical studies using animal models of colitis. However, data covering NGP application in humans are still insufficient, so their safety and proper dosage should be carefully examined in future research. Although FMT appears to be another promising and safe microbial therapy in the treatment of UC, further studies involving larger cohorts will be needed to confirm the optimal dose and route of FMT administration. Finally, targeted administration of postbiotics, different body parts of bacteria, their metabolites, and CFS could have a major advantage over live bacteria due to the minimal risk of infection in patients with a compromised intestinal barrier and immune system. Based on the elucidation and understanding of gut microbial changes, both quantitative and functional, during periods of flare-up and remission, new microbiome-targeted treatment strategies may be effectively developed for UC in the future.

## Figures and Tables

**Figure 1 biomedicines-10-02236-f001:**
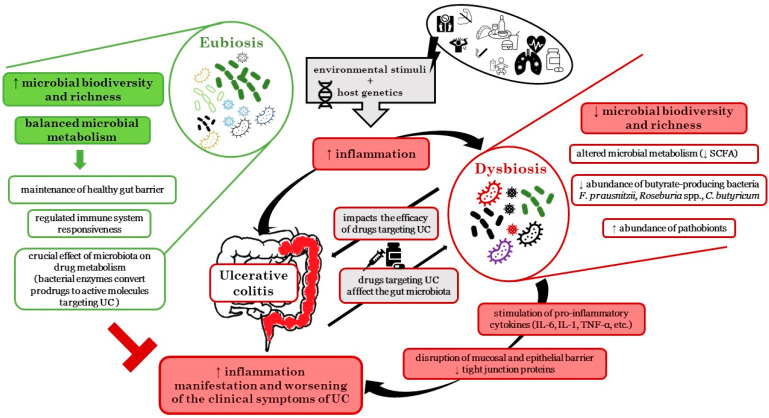
Relationship between UC and development of dysbiosis and vice versa. Abbreviations used: ↑—increased; ↓—decreased; IL—interleukin; SCFA—short chain fatty acids; TNF—tumor necrosis factor; UC—ulcerative colitis.

**Table 1 biomedicines-10-02236-t001:** Efficiency of probiotics with/without prebiotic in UC treatment.

Probiotic Intervention	Concomitant Treatment	Length and Route of Probiotic Application	Trial Type, Stage of Disease, and Sample Size (*n*)	Key Outcomes	Ref.
4 × 10^11^ CFU *Bifidobacterium longum*	12 g/day of fructo-oligosaccharide/inulin mix Synergy 1	4 weeks orally	Double-blind, randomized, controlled trial Active UC *n* = 9 placebo *n* = 9 probiotic	↓ Sigmoidoscopy scores ↓ Inflammation Regeneration of epithelial tissue ↓ Expression of β defensines ↓ TNF-α and IL-1α	[107]
1.8 × 10^10^ CFU *Lacticaseibacillus rhamnosus* GG	2400 mg/day of mesalazine	12 months orally	Prospective, open-label randomized trial Active UC *n* = 65 probiotic *n* = 60 mesalazine *n* = 62 probiotic + mesalazine	LGG prolonged the relapse-free time in comparison with mesalazine	[93]
2.7 × 10^9^ CFU of BIO-THREE *Streptococcus faecalis* T-110, *Clostridium butyricum* TO-A *Bacillus mesentericus* TO-A.	None	4 weeks orally	Mild to moderate distal UC refractory to conventional treatment *n* = 20 probiotic	Improvement of the clinical symptoms and endoscopic findings ↑ Counts of bifidobacteria	[95]
2–3 × 10^11^ CFU *Bifidobacterium longum* BB536	None	24 weeks orally	Open-label study UC refractory to 2250 mg of 5-ASA *n* = 14 probiotic	↓ Clinical activity index, inducing remission skewed the Th1-dominant cytokine profile of splenocytes ↑ Expression of tight junction proteins in colonic mucosa	[94]
3.6 × 10^12^ CFU/day of VSL # 3 *Lacticasebacillus paracasei*, *Lactiplantibacillus plantarum*, *Lactobacillus acidophilus*, *Lactobacillus delbrueckii* subsp. *bulgaricus*, *Bifidobacterium longum*, *Bifidobacterium breve*, *Bifidobacterium*. *infantis*, *Streptococcus thermophilus*	Non-defined stable dose of 5-ASA and/or immunosupressants (azathioprine or 6-mercapropurine)	8 weeks orally	Double-blind, randomized, placebo-controlled study Relapsing mild to moderate UC under treatment with 5-ASA and / or immunosuppressants *n* = 71 probiotic *n* = 73 placebo	↓ Disease activity index	[87]
10^10^ CFU/day *Limosilactobacillus reuteri* ATCC 55730	50 to 75 mg⁄kg⁄day of mesalazine	8 weeks rectally	Prospective, randomized, placebo-controlled study Mild to moderate UC *n* = 20 placebo *n* = 20 probiotic + mesalazine	Rectal infusion of *L. reuteri* improved mucosal inflammation ↑ Mucosal expression levels of IL-10 ↓ Mucosal expression levels of IL-1β, TNF-α, IL-8	[109]
1.6 × 10^9^ CFU/day *Lacticaseibacillus casei* DG	2.4 g/day of 5-ASA	8 weeks orally and rectally	Mild left-sided UC *n* =7 5-ASA *n* = 8 5-ASA + orally probiotic *n* = 11 5-ASA + rectally probiotic	Rectally administered *L. casei* DG ↑ *Lactobacillus* spp. ↓ *Enterobacteriaceae*. ↓ TLR-4 and IL-1β mRNA mucosal expression ↑ Mucosal IL-10 levels	[104]
1.5 × 10^11^ CFU/day of Probio-Tec AB-25 *Lactobacillus acidophilus La-5*, *Bifidobacterium animalis subsp. lactis* BB-12	Non-defined stable dose of 5-ASA	52 weeks orally	Randomized, double-blind, placebo-controlled trial Left-sided UC in remission under monotherapy with 5-ASA *n* = 20 probiotic *n* = 12 placebo	No significant clinical benefit of Probio-Tec AB-25 could be demonstrated in comparison with placebo for maintaining remission.	[101]
1 × 10^10^ CFU/day *Bifidobacterium infantis* 35624	Non-defined optimal dose of mesalazine	6 weeks orally	Randomized, double-blind, placebo-controlled study Mild to moderate active UC *n* = 13 probiotic *n* = 9 placebo	↓ Plasma CRP levels ↓ Systemic pro-inflammatory biomarkers	[103]
3 × 10^9^ CFU/day *Ligilactobacillus salivarius*, *Lactobacillus acidophilus*, *Bifidobacterium bifidum* BGN4	1200 mg;/day of mesalazine	2 years orally	Moderate to severe UC under treatment with mesalazine *n* = 30 mesalazine *n* = 30 mesalazine + probiotic	Improved Mayo Disease Activity Index ↓ Stool frequency Improvement of intestinal mucosa aspect The beneficial effects of probiotics were evident even after two years of treatment.	[96]
3 × 10^9^ CFU *Enterococcus faecium*, *Lactiplantibacillus plantarum*, *Streptococcus thermophilus*, *Bifidobacterium lactis*, *Lactobacillus acidophilus*, *Bifidobacterium longum*	225 mg of fructooligosaccharides	6 weeks orally	Randomized placebo-controlled study. Mild to moderate active UC n = 20 synbiotic n = 20 placebo	↓ CRP and sedimentation values improvement in the clinical activity	[97]
2 × 10^12^ CFU of fermented milk products containing *Bifidobacterium breve* strain Yakult, *Lactobacillus acidophilus*	None	48 weeks orally	Randomized, placebo-controlled, double-blind study Quiescent UC *n* = 98 probiotic *n* = 97 placebo	No effect on time to relapse in UC patients compared with placebo.	[100]
10^13^ CFU of Symprove^TM^ *Lacticaseibacillus rhamnosus* NCIMB 30174, *Lactiplantibacillus plantarum* NCIMB 30173, *Lactobacillus acidophilus* NCIMB 30175, *Enterococcus faecium* NCIMB 30176 i	5-ASA	4 weeks orally .	Randomized, double-blind, placebo-controlled trial UC in remission and on minimal treatment *n* = 40 probiotic *n* = 41 placebo	↓ Fecal calprotectin ↓ Intestinal inflammation Beneficial effect of probiotic supplementation was confirmed only in UC patients, but not in CD patients.	[98]
2 × 10^9^ CFU of Lactocare^®^ *Lacticaseibacillus casei*, *Lactobacillus acidophilus*, *Lacticaseibacillus rhamnosus*, *Lactobacillus bulgaricus*, *Bifidobacterium breve*, *Bifidobacterium longum*, *Streptococcus thermophilus*	Fructooligosaccharide stable dose of mesalazine (at least 1.6 g/day) or 6-mercaptopurine (at least 1 mg/kg/day)	8 weeks orally	Double-blind, semi-randomized, placebo-controlled Mild to moderate active UC *n* = 28 probiotic *n* = 32 placebo	Improvement of gastrointestinal symptoms related to UC; patients with duration of UC for five years or more responded significantly better to Lactocare^®^ treatment than those diagnosed less than five years.	[99]
FEEDColon^®^ *Bifidobacterium bifidum*, *Bifidobacterium lactis*	Fructooligosaccharidecalcium butyrate 5-ASA 2400 mg/day	12 months	Prospective, observational study UC in clinical remission *n* = 21 probiotic *n* = 21 placebo	Improved subjective symptoms (quality of life, abdominal pain, and stool consistency). ↓ Fecal calprotectin 95% patients maintained remission compared to the 57% of those treated with 5-ASA only	[108]

Abbreviations used: ↑—increased; ↓—decreased; 5-ASA—5-aminosalycilates; CD—Crohn disease; CFU—colony forming units; CRP—C reactive protein; IL—interleukin; TNF—tumor necrosis factor; CRP—C reactive protein, TLR—Toll like receptor; UC—ulcerative colitis.

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
