# Peer review of "Probiotic-Based Intervention in the Treatment of Ulcerative Colitis: Conventional and New Approaches"

_biomedicines, 2022, doi:10.3390/biomedicines10092236_

Round 1

Reviewer 1 Report

This a comprehensive review. However it is too long and one may loose the point while reading. Therefore I suggest to shorten and make more briefly the parts that do not concern therapeutic approach and to underline NGP and FMT

Author Response

on behalf of all authors, I thank reviewers for the reviewing the submitted manuscript and for their comments that contributed to increasing the quality of the manuscript.

We used “Track Changes” function as you recommended in an email, all these changes are visible in the reviewed manuscript.

Main changes made to manuscript:

  • Subchapter 3.1 Mechanisms of Probiotic action in UC was shortened
  • Several refferences were deleted (25, 81, 82) and 1 was added (187 – Vernero et al., 2020), therefore the order of the refferences was changed
  • The nomenclature of the genus Lactobacillus was updated

Response to Reviewer 1 Comments

Point 1: This a comprehensive review. However it is too long and one may loose the point while reading. Therefore I suggest to shorten and make more briefly the parts that do not concern therapeutic approach and to underline NGP and FMT

 Response 1: According to your suggestion, the section 3.1 dealing with the mechanisms of action of probiotics in UC was shortened. A detailed description of studies describing the potential mechanisms of individual probiotic strains has been omitted.

Reviewer 2 Report

Journal: Biomedicines (ISSN 2227-9059)

ArticleProbiotic-Based Intervention in the Treatment of Ulcerative Colitis: Conventional and New Approaches, Jana Štofilová *, Monika Kvaková, Anna Kamlárová, Emília Hijová, Izabela Bertkováand Zuzana Guľašová 

Manuscript ID: biomedicines-1863660

This article deals extensively with a current topic, the therapeutic management of patients with inflammatory bowel diseases. The description of the physiopathological mechanisms behind the therapeutic benefits of probiotics or fecal microbiota transplantation in ulcerative colitis increase the value of this article. However, I believe that the information could have been presented more succinctly. 

Author Response

on behalf of all authors, I thank reviewers for the reviewing the submitted manuscript and for their comments that contributed to increasing the quality of the manuscript.

We used “Track Changes” function as you recommended in an email, all these changes are visible in the reviewed manuscript.

Main changes made to manuscript:

  • Subchapter 3.1 Mechanisms of Probiotic action in UC was shortened
  • Several refferences were deleted (25, 81, 82) and 1 was added (187 – Vernero et al., 2020), therefore the order of the refferences was changed
  • The nomenclature of the genus Lactobacillus was updated

Response to Reviewer 2 Comments

Point 1: This article deals extensively with a current topic, the therapeutic management of patients with inflammatory bowel diseases. The description of the physiopathological mechanisms behind the therapeutic benefits of probiotics or fecal microbiota transplantation in ulcerative colitis increase the value of this article. However, I believe that the information could have been presented more succinctly.

Response 1: Since there are dozens of studies describing the possible mechanism of action of probiotics in animal models of colitis, we have highlighted at least some relevant to the topic in the manuscript. However, we agree with the reviewer that this section is not key to the issue and therefore, according to your proposal, the section 3.1 Mechanisms of action of probiotics in UC was shortened.

Reviewer 3 Report

-          “Inflammatory bowel disease (IBD) has become a global disease with accelerating incidence in industrialized countries whose societies have become more westernized”

Incidence of IBD is stable in industrialized countries while it is increasing in developing countries

-          Use oxford comma in the whole text

-          Define the abbreviation (for example, F. prausnitzii)

-          “The number of SCFA producing bacteria such as …. and F. prausnitzii was reported to be decreased in patients with UC” “Furthermore, closer analysis of the gut microbiota at the genus and species level pointed out that UC is associated with significant increase …. F. prausnitzii”

Decrease or increase ?!

-          Add small molecules among UC therapies

-          Explain better the “immunosuppressive properties” of ASAs

-          “The efficacy of sulfasalazine, the drug of the first choice for UC”

Maybe decades ago

-          Cite “The Usefulness of Microencapsulated Sodium Butyrate Add-On Therapy in Maintaining Remission in Patients with Ulcerative Colitis: A Prospective Observational Study. J Clin Med. 2020 Dec 4;9(12):3941. doi: 10.3390/jcm9123941. PMID: 33291846; PMCID: PMC7762036.” 

Author Response

on behalf of all authors, I thank reviewers for the reviewing the submitted manuscript and for their comments that contributed to increasing the quality of the manuscript.

We used “Track Changes” function as you recommended in an email, all these changes are visible in the reviewed manuscript.

Main changes made to manuscript:

  • Subchapter 3.1 Mechanisms of Probiotic action in UC was shortened
  • Several refferences were deleted (25, 81, 82) and 1 was added (187 – Vernero et al., 2020), therefore the order of the refferences was changed
  • The nomenclature of the genus Lactobacillus was updated

Response to Reviewer 3 Comments

Point 1: “Inflammatory bowel disease (IBD) has become a global disease with accelerating incidence in industrialized countries whose societies have become more westernized”

Incidence of IBD is stable in industrialized countries while it is increasing in developing countries

Response 2: Thank you for your valuable comment. It was meant in "newly" industrialized countries according to a study by Ng et al., 2017 (Ref. 2). The word "newly" was added to the text.

Point 2: Use oxford comma in the whole text

Response 2: The whole text was carefully examined and missing commas have been added.

 Point 3: Define the abbreviation (for example, F. prausnitzii)

Response 3: The whole text was thoroughly checked and abbreviations were defined at the first mention in the text. Latin names of microorganisms were abbreviated using a generally accepted rule. Once the complete name of a microorganism was written out once, the genus name was abbreviated to just the capital letter, so that there would be no confusion with other genera (e.g. A. muciniphila, F. prausnitzii, L. plantarum, etc.).

Point 4: “The number of SCFA producing bacteria such as …. and F. prausnitzii was reported to be decreased in patients with UC” “Furthermore, closer analysis of the gut microbiota at the genus and species level pointed out that UC is associated with significant increase …. F. prausnitzii”

Decrease or increase ?!

Response 4: According to a study by Gryaznova et al. (2021), F. prausnitzi was indeed elevated in UC patients. The authors attribute this discrepancy with the literature to population peculiarities of microbiome composition, as the study was conducted on a Russian population. Furthermore, the study included only 10 UC patients (according to Fig. 3, only 3 UC patients actually had a higher abundance of F. prausnitzii) and 10 controls, which represents a statistically small subset. After detailed analysis of the given study, we decided to remove this claim from the manuscript.

Point 5: Add small molecules among UC therapies

Response 5: Small molecules were added among UC therapies.

Point 6: Explain better the “immunosuppressive properties” of ASAs.

Response 6: Thank you for pointing my mistake out. ASA is definitely not an immunosuppressant, they have anti-inflammatory properties. The word "immunosuppressive" was deleted.

Point 7: “The efficacy of sulfasalazine, the drug of the first choice for UC”

Maybe decades ago

Response 7. Thank you for pointing my mistake out. We agree with reviewer and this part was deleted.

Point 8: Cite “The Usefulness of Microencapsulated Sodium Butyrate Add-On Therapy in Maintaining Remission in Patients with Ulcerative Colitis: A Prospective Observational Study. J Clin Med. 2020 Dec 4;9(12):3941. doi: 10.3390/jcm9123941. PMID: 33291846; PMCID: PMC7762036.”

Response 8: Relevant results of the study Vernero. et al. 2020 was added to the manuscript (ref 187).